# Prediction of Spatial Point Processes: Regularized Method with Out-of-Sample Guarantees

**Muhammad Osama***
muhammad.osama@it.uu.se

**Dave Zachariah***
dave.zachariah@it.uu.se

**Peter Stoica***
peter.stoica@it.uu.se

*Division of System and Control, Department of Information Technology, Uppsala University

## Abstract

A spatial point process can be characterized by an intensity function which predicts the number of events that occur across space. In this paper, we develop a method to infer predictive intensity intervals by learning a spatial model using a regularized criterion. We prove that the proposed method exhibits out-of-sample prediction performance guarantees which, unlike standard estimators, are valid even when the spatial model is misspecified. The method is demonstrated using synthetic as well as real spatial data.

## 1 Introduction

Spatial point processes can be found in a range of applications from astronomy and biology to ecology and criminology. These processes can be characterized by a nonnegative intensity function $\lambda(\boldsymbol{x})$ which predicts the number of events that occur across space parameterized by $\boldsymbol{x} \in \mathcal{X}$ [8, 4].

A standard approach to estimate the intensity function of a process is to use nonparametric kernel density-based methods [6, 7]. These smoothing techniques require, however, careful tuning of kernel bandwidth parameters and are, more importantly, subject to selection biases. That is, in regions where no events have been observed, the intensity is inferred to be zero and no measure is readily available for a user to assess the uncertainty of such predictions. More advanced methods infer the intensity by assuming a parameterized model of the data-generating process, such as inhomogeneous Poisson point process models. One popular model is the log-Gaussian Cox process (LGCP) model [9] where the intensity function is modeled as a Gaussian process [22] via a logarithmic link function to ensure non-negativity. However, the infinite dimensionality of the intensity function makes this model computationally prohibitive and substantial effort has been devoted to develop more tractable approximation methods based on gridding [9, 13], variational inference [15, 12], Markov chain Monte Carlo [2] and Laplace approximations [20] for the log and other link functions. A more fundamental problem remains in that their resulting uncertainty measures are not calibrated to the actual out-of-sample variability of the number of events across space. Poor calibration consequently leads to unreliable inferences of the process.

In this paper, we develop a spatially varying intensity interval with provable out-of-sample performance guarantees. Our contributions can be summarized as follows:

- the interval reliably covers out-of-sample events with a specified probability by building on the conformal prediction framework [19],

- it is constructed using a predictive spatial Poisson model with provable out-of-sample accuracy,

- its size appropriately increases in regions with missing data to reflect inherent uncertainty and mitigate sampling biases,

- the statistical guarantees remain valid even when the assumed Poisson model is misspecified.

Thus the proposed method yields both reliable and informative predictive intervals under a wider range of conditions than standard methods which depend on the assumed model, e.g. LGCP [9], to match the unknown data-generating process.

*Notations:* $\mathbb{E}_n[a] = n^{-1}\sum_{i=1}^{n} a_i$ denotes the sample mean of $a$. The element-wise Hadamard product is denoted $\odot$.

## 2 Problem formulation

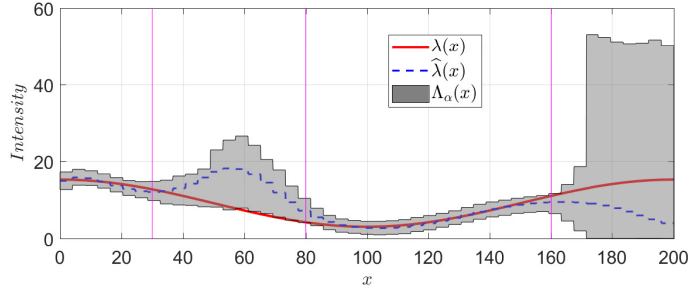

Figure 1: Unknown intensity function $\lambda(x)$ (solid) expressed in number of counts per unit of area, across a one-dimensional spatial domain $\mathcal{X} = [0, 200]$ which is discretized into 50 regions. Intensity interval $\Lambda_\alpha(x)$ with $1 - \alpha = 80\%$ out-of-sample coverage (3) inferred using $n = 50$ samples. Estimated intensity function $\widehat{\lambda}(\boldsymbol{x})$ (dashed). Data is missing in the regions $[30, 80]$ and $[160, 200]$ where the intensity interval increases appropriately.

The intensity function $\lambda(\boldsymbol{x})$ of a spatial process is expressed as the number of events per unit area and varies over a spatial domain of interest, $\mathcal{X}$, which we equipartition into $R$ disjoint regions: $\mathcal{X} = \bigcup_{r=1}^{R} \mathcal{X}_r \subset \mathbb{R}^d$ and is a common means of modelling continuous inhomogeneous point processes, see [9, 13]. The function $\lambda(\boldsymbol{x})$ determines the expected number of events $y \in \{0, \dots, Y\}$ that occur in region $\mathcal{X}_r$ by

$$\mathbb{E}[y|r] = \int_{\mathcal{X}_r} \lambda(\boldsymbol{x})d\boldsymbol{x}, \tag{1}$$

where $r$ is the region index and $Y$ is the maximum number of counts.

We observe $n$ independent samples drawn from the process,

$$(r_i, y_i) \sim p(r)p(y|r), \tag{2}$$

where the data-generating distribution is unknown. Let the collection of pairwise datapoints be denoted $(\boldsymbol{r}, \boldsymbol{y}) = \{(r_1, y_1), \dots, (r_n, y_n)\}$. Given this dataset, our goal is to infer an intensity interval $\Lambda(\boldsymbol{x}) \subset [0, \infty)$ of the unknown spatial point process, which predicts the number of events per unit area at location $\boldsymbol{x}$. See Figure 1 for an illustration in one-dimensional space. A reliable interval $\Lambda_\alpha(\boldsymbol{x})$ will cover a new out-of-sample observation $y$ in a region $r$ with a probability of at least $1 - \alpha$. That is, for a specified level $\alpha$ the out-of-sample coverage is

$$\Pr\left\{ y \in \Lambda_\alpha(\boldsymbol{x})|\mathcal{X}_r|, \ \forall \boldsymbol{x} \in \mathcal{X}_r \right\} \geqslant 1 - \alpha, \tag{3}$$

where $|\mathcal{X}_r|$ is the area of the $r$th region. Since the trivial noninformative interval $[0, \infty)$ also satisfies (3), our goal is to construct $\Lambda_\alpha(\boldsymbol{x})$ that is both reliable and informative.

# 3 Inference method

We begin by showing that an intensity interval $\Lambda_\alpha(\boldsymbol{x})$ with reliable out-of-sample coverage can be constructed using the conformal prediction framework [19]. Note that obtaining tractable and informative intervals in this approach requires learning an accurate predictor in a computationally efficient manner. We develop such a predictor and prove that it has finite-sample and distribution-free performance guarantees. These guarantees are independent of the manner in which space is discretized.

## 3.1 Conformal intensity intervals

Let $\mathbb{E}_{\boldsymbol{\theta}}[y|r]$ denote a predictor parameterized by a vector $\boldsymbol{\theta}$. For a given region $r$, consider a new data point $(r, \widetilde{y})$, where $\widetilde{y}$ represents number of counts and takes a value between $[0, Y]$. The principle of conformal prediction is to quantify how well this new point conforms to the observed data $(\boldsymbol{r}, \boldsymbol{y})$. This is done by first fitting parameters $\boldsymbol{\theta}'$ to the augmented set $(\boldsymbol{r}, \boldsymbol{y}) \cup (r, \widetilde{y})$ and then using the score

$$\pi(\widetilde{y}) = \frac{1}{n+1} \sum_{i=1}^{n+1} \mathcal{I}\Big(e_i \leqslant \big|\widetilde{y} - \mathbb{E}_{\boldsymbol{\theta}'}[y|r]\big|\Big) \in (0, 1], \tag{4}$$

where $\mathcal{I}(\cdot)$ is the indicator function and $e_i = |y_i - \mathbb{E}_{\boldsymbol{\theta}'}[y|r_i]|$ are residuals for all observed data points $i = 1, \ldots, n$. When a new residual $\big|\widetilde{y} - \mathbb{E}_{\boldsymbol{\theta}'}[y|r]\big|$ is statistically indistinguishable from the rest, $\pi(\widetilde{y})$ corresponds to a p-value [19]. On this basis we construct an intensity interval $\Lambda_\alpha(\boldsymbol{x})$ by including all points $\widetilde{y}$ that conform to the dataset with significance level $\alpha$, as summarized in Algorithm 1. Using [14, thm. 2.1], we can prove that $\Lambda_\alpha(\boldsymbol{x})$ satisfies the out-of-sample coverage (3).

---

**Algorithm 1** Conformal intensity interval

1: **Input**: Location $\boldsymbol{x}$, significance level $\alpha$, data $(\boldsymbol{r}, \boldsymbol{y})$
2: **for** all $\widetilde{y} \in \{0, \ldots, Y\}$ **do**
3:     Set $r$ if $\boldsymbol{x} \in \mathcal{X}_r$
4:     Update predictor $\mathbb{E}_{\boldsymbol{\theta}}[y|r]$ using augmented data $(\boldsymbol{r}, \boldsymbol{y}) \cup (r, \widetilde{y})$
5:     Compute score $\pi(\widetilde{y})$ in (4)
6: **end for**
7: **Output**: $\Lambda_\alpha(\boldsymbol{x}) = \{\widetilde{y} : (n+1)\pi(\widetilde{y}) \leqslant \lceil (n+1)\alpha \rceil\}/|\mathcal{X}_r|$

---

While this approach yields reliable out-of-sample coverage guarantees, there are two possible limitations:

1. The residuals can be decomposed as $e = (\mathbb{E}[y|r] - \mathbb{E}_{\boldsymbol{\theta}}[y|r]) + \varepsilon$, where the term in brackets is the model approximation error and $\varepsilon$ is an irreducible zero-mean error. Obtaining informative $\Lambda_\alpha(\boldsymbol{x})$ across space requires learned predictors with small model approximation errors.

2. Learning methods that are computationally demanding render the computation of $\Lambda_\alpha(\boldsymbol{x})$ intractable across space, since the conformal method requires re-fitting the predictor multiple times for each region.

Next, we focus on addressing both limitations.

## 3.2 Spatial model

We seek an accurate model $p_{\boldsymbol{\theta}}(y|r)$ of $p(y|r)$, parameterized by $\boldsymbol{\theta}$. For a given $\boldsymbol{r}$, we quantify the out-of-sample accuracy of a model by the Kullback-Leibler divergence *per sample*,

$$\mathcal{R}(\boldsymbol{\theta}) = \frac{1}{n} \mathbb{E}_{\boldsymbol{y}|\boldsymbol{r}} \left[ \ln \frac{p(\boldsymbol{y}|\boldsymbol{r})}{p_{\boldsymbol{\theta}}(\boldsymbol{y}|\boldsymbol{r})} \right] \geqslant 0, \quad \text{for which} \quad \mathcal{R}(\boldsymbol{\theta}) = 0 \Leftrightarrow p_{\boldsymbol{\theta}}(\boldsymbol{y}|\boldsymbol{r}) \equiv p(\boldsymbol{y}|\boldsymbol{r}) \tag{5}$$

In general, the unknown intensity function underlying $p(y|r)$ has a local spatial structure and can be modeled as smooth since we expect counts in neighbouring regions to be similar in real-world

applications. On this basis, we consider following the class of models,

$$\mathcal{P}_{\boldsymbol{\theta}} = \left\{ p_{\boldsymbol{\theta}}(y|r) \text{ is Poisson with mean } \mathbb{E}_{\boldsymbol{\theta}}[y|r] = \exp(\boldsymbol{\phi}^\top(r)\boldsymbol{\theta}), \; \boldsymbol{\theta} \in \mathbb{R}^R \right\},$$

where $\boldsymbol{\phi}(r)$ is $R \times 1$ spatial basis vector whose components are given by the cubic b-spline function [21] (see supplementary material). The Poisson distribution is the maximum entropy distribution for count data and is here parameterized via a latent field $\{\theta_1, \ldots, \theta_R\}$ across regions [4, ch. 4.3]. Using a cubic b-spline basis [21], we model the mean in region $r$ via a weighted average $\boldsymbol{\phi}(r)^\top \boldsymbol{\theta}$ of latent parameters from neighbouring regions, where the maximum weight in $\boldsymbol{\phi}(r)$ is less than 1. This parameterization yields locally smooth spatial structures and is similar to using a latent process model for the mean as in the commonly used LGCP model [9, sec. 4.1].

The unknown optimal predictive Poisson model is given by

$$\boldsymbol{\theta}^\star = \arg\min_{\boldsymbol{\theta}} \; \mathcal{R}(\boldsymbol{\theta}) \tag{6}$$

and has an out-of-sample accuracy $\mathcal{R}(\boldsymbol{\theta}^\star)$.

## 3.3 Regularized learning criterion

We propose learning a spatial Poisson model in $\mathcal{P}_{\boldsymbol{\theta}}$ using the following learning criterion

$$\widehat{\boldsymbol{\theta}} = \arg\min_{\boldsymbol{\theta}} \; -n^{-1} \ln p_{\boldsymbol{\theta}}(\boldsymbol{y}|\boldsymbol{r}) \; + \; n^{-\gamma}||\boldsymbol{w} \odot \boldsymbol{\theta}||_1, \tag{7}$$

where $\ln p_{\boldsymbol{\theta}}(\boldsymbol{y}|\boldsymbol{r})$ is the log-likelihood, which is convex [18], and $\boldsymbol{w}$ is a given vector of regularization weights. The regularization term in (7) not only mitigates overfitting of the model by penalizing parameters in $\boldsymbol{\theta}$ individually, it also yields the following finite sample and distribution-free result.

**Theorem 1** *Let $\gamma \in (0, \frac{1}{2})$, then the out-of-sample accuracy of the learned model is bounded as*

$$\boxed{\mathcal{R}(\widehat{\boldsymbol{\theta}}) \leqslant \mathcal{R}(\boldsymbol{\theta}^\star) + 2n^{-\gamma}||\boldsymbol{w} \odot \boldsymbol{\theta}^\star||_1} \tag{8}$$

*with a probability of at least*

$$\max\left(0, \, 1 - 2R\exp\left\{-\frac{w_o^2 n^{1-2\gamma}}{2Y^2}\right\}\right), \quad where \quad w_o = \min_{k=1,\ldots,R} w_k.$$

We provide an outline of the proof in Section 3.3.1, while relegating the details to the Supplementary Material. The above theorem guarantees that the out-of-sample accuracy $\mathcal{R}(\widehat{\boldsymbol{\theta}})$ of the learned model (7) will be close to $\mathcal{R}(\boldsymbol{\theta}^\star)$ of the optimal model (6), even if the model class (3.2) does not contain the true data-generating process. As $\gamma$ is increased, the bound tightens and the probabilistic guarantee weakens, but for a given data set one can readily search for the value of $\gamma \in (0, 0.5)$ which yields the most informative interval $\Lambda_\alpha(\boldsymbol{x})$.

The first term of (7) contains inner products $\boldsymbol{\phi}^\top(r)\boldsymbol{\theta}$ which are formed using a regressor matrix. To balance fitting with the regularizing term in (7), it is common to rescale all columns of the regressor matrix to unit norm. An equivalent way is to choose the following regularization weights $w_k = \sqrt{\mathbb{E}_n[|\phi_k(r)|^2]}$, see e.g. [3]. We then obtain a predictor as

$$\mathbb{E}_{\widehat{\boldsymbol{\theta}}}[y|r] = \exp(\boldsymbol{\phi}^\top(r)\widehat{\boldsymbol{\theta}})$$

and predictive intensity interval $\Lambda_\alpha(\boldsymbol{x})$ via Algorithm 1. Setting $w_k \equiv 0$ in (7) yields a maximum likelihood model with less informative intervals, as we show in the numerical experiments section.

### 3.3.1 Proof of theorem

The minimizer $\widehat{\boldsymbol{\theta}}$ in (7) satisfies

$$\widehat{R}(\widehat{\boldsymbol{\theta}}) \leqslant \widehat{R}(\boldsymbol{\theta}^\star) + \rho f(\boldsymbol{\theta}^\star) - \rho f(\widehat{\boldsymbol{\theta}}), \tag{9}$$

where $\widehat{R}(\boldsymbol{\theta}) = n^{-1} \ln \frac{p(\boldsymbol{y}|\boldsymbol{r})}{p_{\boldsymbol{\theta}}(\boldsymbol{y}|\boldsymbol{r})}$ is the in-sample divergence, corresponding to (5), $f(\boldsymbol{\theta}) = ||\boldsymbol{w} \odot \boldsymbol{\theta}||_1$ and $\rho = n^{-\gamma}$.

Using the functional form of the Poisson distribution, we have

$$-\ln p_{\boldsymbol{\theta}}(\boldsymbol{y}|\boldsymbol{r}) \;=\; \sum_{i=1}^{n} -\ln p_{\boldsymbol{\theta}}(y_i|r_i) \;=\; \sum_{i=1}^{n} \mathbb{E}_{\boldsymbol{\theta}}[y_i|r_i] \;-\; y_i \ln(\mathbb{E}_{\boldsymbol{\theta}}[y_i|r_i]) \;+\; \ln(y_i!)$$

Then the gap between the out-of-sample and in-sample divergences for any given model $\boldsymbol{\theta}$ is given by

$$\begin{aligned}
\mathcal{R}(\boldsymbol{\theta}) - \widehat{R}(\boldsymbol{\theta}) &= \frac{1}{n}\Big[ \ln p_{\boldsymbol{\theta}}(\boldsymbol{y}|\boldsymbol{r}) - \mathbb{E}_{\boldsymbol{y}|\boldsymbol{r}}[\ln p_{\boldsymbol{\theta}}(\boldsymbol{y}|\boldsymbol{r})] + \mathbb{E}_{\boldsymbol{y}|\boldsymbol{r}}[\ln p(\boldsymbol{y}|\boldsymbol{r})] - \ln p(\boldsymbol{y}|\boldsymbol{r}) \Big] \\
&= \mathbb{E}_n\Big[ (y - \mathbb{E}_{y|r}[y])\boldsymbol{\phi}(r) \Big]^{\top} \boldsymbol{\theta} + \frac{1}{n} K,
\end{aligned} \tag{10}$$

where the second line follows from using our Poisson model $\mathcal{P}_{\boldsymbol{\theta}}$ and $K = \mathbb{E}_{\boldsymbol{y}|\boldsymbol{r}}[\ln p(\boldsymbol{y}|\boldsymbol{r})] - \ln p(\boldsymbol{y}|\boldsymbol{r}) + \sum_{i=1}^{n} \mathbb{E}_{\boldsymbol{y}|\boldsymbol{r}}[\ln(y_i!)] - \ln(y_i!)$ is a constant. Note that the divergence gap is linear in $\boldsymbol{\theta}$, and we can therefore relate the gaps for the optimal model $\widehat{\boldsymbol{\theta}}$ with the learned model $\boldsymbol{\theta}^{\star}$ as follows:

$$\big[\mathcal{R}(\boldsymbol{\theta}^{\star}) - \widehat{R}(\boldsymbol{\theta}^{\star})\big] - \big[\mathcal{R}(\widehat{\boldsymbol{\theta}}) - \widehat{R}(\widehat{\boldsymbol{\theta}})\big] = \boldsymbol{g}^{\top}(\boldsymbol{\theta}^{\star} - \widehat{\boldsymbol{\theta}}), \tag{11}$$

where

$$\boldsymbol{g} \equiv \partial_{\boldsymbol{\theta}}[\mathcal{R}(\boldsymbol{\theta}) - \widehat{R}(\boldsymbol{\theta})]\big|_{\boldsymbol{\theta}=\widehat{\boldsymbol{\theta}}} = \Big[ \mathbb{E}_n[z_1], \ldots, \mathbb{E}_n[z_R] \Big]^{\top},$$

is the gradient of (10) and we introduce the random variable $z_k = (y - \mathbb{E}_{y|r}[y])\phi_k(r) \in [-Y, Y]$ for notational simplicity (see supplementary material).

Inserting (9) into (11) and re-arranging yields

$$\mathcal{R}(\widehat{\boldsymbol{\theta}}) \leqslant \mathcal{R}(\boldsymbol{\theta}^{\star}) - \boldsymbol{g}^{\top}(\boldsymbol{\theta}^{\star} - \widehat{\boldsymbol{\theta}}) + \rho f(\boldsymbol{\theta}^{\star}) - \rho f(\widehat{\boldsymbol{\theta}}), \tag{12}$$

where the RHS is dependent on $\widehat{\boldsymbol{\theta}}$. Next, we upper bound the RHS by a constant that is independent of $\widehat{\boldsymbol{\theta}}$.

The weighted norm $f(\boldsymbol{\theta})$ has an associated dual norm

$$\widetilde{f}(\boldsymbol{g}) = \sup_{\boldsymbol{\theta}:f(\boldsymbol{\theta})\leqslant 1} \boldsymbol{g}^{\top}\boldsymbol{\theta} \equiv \frac{||\boldsymbol{g}||_{\infty}}{w_o} = \max_{k=1,\ldots,R} \frac{|\mathbb{E}_n[z_k]|}{w_o}$$

see the supplementary material. Using the dual norm, we have the following inequalities

$$-\boldsymbol{g}^{\top}\boldsymbol{\theta}^{\star} \leqslant \widetilde{f}(\boldsymbol{g})f(\boldsymbol{\theta}^{\star}) \quad \text{and} \quad \boldsymbol{g}^{\top}\widehat{\boldsymbol{\theta}} \leqslant \widetilde{f}(\boldsymbol{g})f(\widehat{\boldsymbol{\theta}})$$

and combining them with (12), as in [23], yields

$$\mathcal{R}(\widehat{\boldsymbol{\theta}}) \leqslant \mathcal{R}(\boldsymbol{\theta}^{\star}) + (\rho + \widetilde{f}(\boldsymbol{g}))f(\boldsymbol{\theta}^{\star}) + (\widetilde{f}(\boldsymbol{g}) - \rho)f(\widehat{\boldsymbol{\theta}}) \leqslant \mathcal{R}(\boldsymbol{\theta}^{\star}) + 2\rho f(\boldsymbol{\theta}^{\star}) \tag{13}$$

when $\rho \geqslant \widetilde{f}(\boldsymbol{g})$. The probability of this event is lower bounded by

$$\Pr\big(\rho \geqslant \widetilde{f}(\boldsymbol{g})\big) \geqslant 1 - 2R\exp\Big[ -\frac{w_o^2 n^{1-2\gamma}}{2Y^2} \Big] \tag{14}$$

We derive this bound using Hoeffding's inequality, for which

$$\Pr(|\mathbb{E}_n[z_k] - \mathbb{E}[z_k]| \leqslant \epsilon) \geqslant 1 - 2\exp\Big[ -\frac{n\epsilon^2}{2Y^2} \Big], \tag{15}$$

and $\mathbb{E}[z_k] = \mathbb{E}_r\big[ (\mathbb{E}_{y|r}[y] - \mathbb{E}_{y|r}[y])\phi_k(r) \big] = 0$. Moreover,

$$\Pr\Big( \max_{k=1,\ldots,R} |\mathbb{E}_n[z_k]| \leqslant \epsilon \Big) = \Pr\Big( \bigcap_{k=1}^{R} |\mathbb{E}_n[z_k]| \leqslant \epsilon \Big) \geqslant 1 - 2R\exp\Big[ -\frac{n\epsilon^2}{2Y^2} \Big],$$

using DeMorgan's law and the union bound (see supplementary material). Setting $\epsilon = w_o\rho$, we obtain (14) Hence equation (13) and (14) prove Theorem 1. It can be seen that for $\gamma \in (0, \frac{1}{2})$, the probability bound on the right hand side increases with $n$.

### 3.3.2 Minimization algorithm

To solve the convex minimization problem (7) in a computationally efficient manner, we use a majorization-minimization (MM) algorithm. Specifically, let $V(\boldsymbol{\theta}) = -n^{-1} \ln p_{\boldsymbol{\theta}}(\boldsymbol{y}|\boldsymbol{r})$ and $f(\boldsymbol{\theta}) = ||\boldsymbol{w} \odot \boldsymbol{\theta}||_1$ then we bound the objective in (7) as

$$V(\boldsymbol{\theta}) + n^{-\gamma} f(\boldsymbol{\theta}) \leqslant Q(\boldsymbol{\theta}; \widetilde{\boldsymbol{\theta}}) + n^{-\gamma} f(\boldsymbol{\theta}), \tag{16}$$

where $Q(\boldsymbol{\theta}; \widetilde{\boldsymbol{\theta}})$ is a quadratic majorizing function of $V(\boldsymbol{\theta})$ such that $Q(\widetilde{\boldsymbol{\theta}}; \widetilde{\boldsymbol{\theta}}) = V(\widetilde{\boldsymbol{\theta}})$, see [18, ch. 5]. Minimizing the right hand side of (16) takes the form of a weighted lasso regression and can therefore be solved efficiently using coordinate descent. The pseudo-code is given in Algorithm 2, see the supplementary material for details. The runtime of Algorithm 2 scales as $O(nR^2)$ i.e. linear in number of datapoints $n$. This computational efficiency of Algorithm 2 is leveraged in Algorithm 1 when updating the predictor $\mathbb{E}_{\boldsymbol{\theta}}[y|r]$ with an augmented dataset $(\boldsymbol{r}, \boldsymbol{y}) \cup (r, \widetilde{y})$. This renders the computation of $\Lambda_\alpha(\boldsymbol{x})$ tractable across space.

---

**Algorithm 2** Majorization-minimization method

1: **Input**: Data $(\boldsymbol{r}, \boldsymbol{y})$, parameter $\gamma \in (0, \frac{1}{2})$ and $Y$
2: Form weights $w_k = \sqrt{\mathbb{E}_n[|\phi_k(r)|^2]}$ for $k = 1, \ldots, R$
3: Set $\widetilde{\boldsymbol{\theta}} := \boldsymbol{0}$
4: **while**
5: Form quadratic approximation at $\widetilde{\boldsymbol{\theta}}$: $Q(\boldsymbol{\theta}; \widetilde{\boldsymbol{\theta}}) + n^{-\gamma} ||\boldsymbol{w} \odot \boldsymbol{\theta}||_1$
6: Solve $\check{\boldsymbol{\theta}} := \arg\min_{\boldsymbol{\theta}} Q(\boldsymbol{\theta}; \widetilde{\boldsymbol{\theta}}) + n^{-\gamma} ||\boldsymbol{w} \odot \boldsymbol{\theta}||_1$ using coordinate descent
7: $\widetilde{\boldsymbol{\theta}} := \check{\boldsymbol{\theta}}$
8: **until** $||\widehat{\boldsymbol{\theta}} - \check{\boldsymbol{\theta}}|| \geqslant \epsilon$
9: **Output**: $\widehat{\boldsymbol{\theta}} = \check{\boldsymbol{\theta}}$ and $\mathbb{E}_{\widehat{\boldsymbol{\theta}}}[y|r] = \exp(\boldsymbol{\phi}^\top(r)\widehat{\boldsymbol{\theta}})$

---

The code for algorithms 1 and 2 are provided on github.

## 4 Numerical experiments

We demonstrate the proposed method using both synthetic and real spatial data.

### 4.1 Synthetic data with missing regions

To illustrate the performance of our learning criterion in (7), we begin by considering a one-dimensional spatial domain $\mathcal{X} = [0, 100]$, equipartitioned into $R = 20$ regions. Throughout we use $\gamma = 0.499$ in (7).

**Comparison with log-Gaussian Cox process model**

We consider a process described by the intensity function

$$\lambda(x) = 10 \exp\left(-\frac{x}{50}\right), \tag{17}$$

and sample events using a spatial Poisson process model using inversion sampling [5]. The distribution $p(y|r)$ is then Poisson. Using a realization $(\boldsymbol{r}, \boldsymbol{y})$, we compare our predictive intensity interval $\Lambda_\alpha(x)$ with a $(1-\alpha)\%$-credibility interval $\widetilde{\Lambda}_\alpha(x)$ obtained by assuming an LGCP model for the $\lambda(x)$ [9] and approximating its posterior belief distribution using integrated nested Laplace approximation (INLA) [17, 11]. For the cubic b-splines in $\mathcal{P}_{\boldsymbol{\theta}}$, the spatial support of the weights in $\phi(r)$ was set to cover all regions.

We consider interpolation and extrapolation cases where the data is missing across $[30, 80]$ and $[70, 100]$, respectively. Figures 2a and 2b show the intervals both cases. While $\widetilde{\Lambda}_\alpha(x)$ is tighter than $\Lambda_\alpha(x)$ in the missing data regions, it has no out-of-sample guarantees and therefore lacks reliability. This is critically evident in the extrapolation case, where $\Lambda_\alpha(x)$ becomes noninformative further away from the observed data regions. By contrast, $\widetilde{\Lambda}_\alpha(x)$ provides misleading inferences in this case.

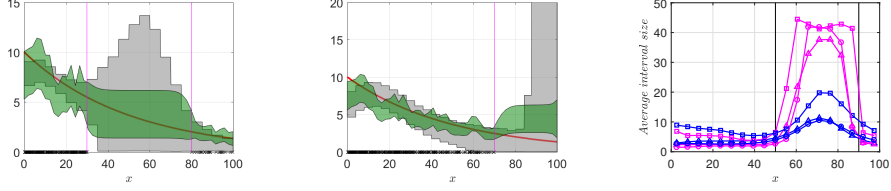

(a) Interpolation with data miss- (b) Extrapolation with data miss- (c) Average interval size with data
ing in $[30, 80]$             ing in $[70, 100]$             missing in $[50, 90]$

Figure 2: (a) Interpolation and (b) extrapolation with $\Lambda_\alpha(x)$ (grey) and $\widetilde{\Lambda}_\alpha(x)$ (green) with $1 - \alpha = 80\%$, for a given realization of point data (black crosses). The unknown intensity function $\lambda(x)$ (red) gives the expected number of events in a region, see (1). (c) Misspecified case with average intensity interval size $|\Lambda_\alpha(x)|$, using nonzero (blue) and zero (red) regularization weights in (7). Data in $[50, 90]$ is missing. The different markers correspond to three different spatial processes, with intensity functions $\lambda_1(x)$, $\lambda_2(x)$ and $\lambda_3(x)$. The out-of-sample coverage (3) was set to be at least $1 - \alpha = 80\%$ and the empirical coverage is given in 1.

| Empirical coverage of $\Lambda_\alpha(x)$ [%] | | |
|---|---|---|
| $\alpha = 0.2$ | Proposed | Unregularized |
| $\lambda_1$ | 97.05 | 97.37 |
| $\lambda_2$ | 91.05 | 98.32 |
| $\lambda_3$ | 81.37 | 95.37 |

Table 1: Comparison of empirical coverage of $\Lambda_\alpha(x)$, using the proposed regularized vs. the unregularized maximum likelihood method. We target $\geqslant 1 - \alpha = 80\%$ coverage.

**Comparison with unregularized maximum likelihood model**

Next, we consider a three different spatial processes, described by intensity functions

$$\lambda_1(x) = \frac{500}{\sqrt{2\pi 25^2}} \exp\Big[ - \frac{(x - 50)^2}{2 \times 25^2}\Big], \; \lambda_2(x) = 5 \sin(\frac{2\pi}{50}x) + 5, \; \lambda_3(x) = \frac{3}{8}\sqrt{x}.$$

For the first process, the intensity peaks at $x = 50$, the second process is periodic with a period of 50 spatial units, and for the third process the intensity grows monotonically with space $x$. In all three cases, the number of events in a given region is then drawn as $y \sim p(y|r)$ using a negative binomial distribution, with mean given by (1) and number of failures set to 100, yielding a dataset $(r, y)$. Note that the Poisson model class $\mathcal{P}_\theta$ is misspecified here.

We set the nominal out-of-sample coverage $\geqslant 80\%$ and compare the interval sizes $|\Lambda_\alpha(x)|$ across space and the overall empirical coverage, when using regularized and unregularized criteria (7), respectively. The averages are formed using 50 Monte Carlo simulations.

Figure 2c and Table 1 summarize the results of comparison between the regularized and unregularized approaches for the three spatial processes. While both intervals exhibit the same out-of-sample coverage (table 1), the unregularized method results in intervals that are nearly four times larger than those of the proposed method (figure 2c) in the missing region.

## 4.2 Real data

In this section we demonstrate the proposed method using two real spatial data sets. In two-dimensional space it is challenging to illustrate a varying interval $\Lambda_\alpha(\boldsymbol{x})$, so for clarity we show its maximum value, minimium value and size as well as compare it with a point estimate obtained from the predictor, i.e.,

$$\widehat{\lambda}(\boldsymbol{x}) = \sum_{r=1}^{R} \mathcal{I}(\boldsymbol{x} \in \mathcal{X}_r)\frac{\mathbb{E}_{\widehat{\boldsymbol{\theta}}}[y|r]}{|\mathcal{X}_r|} \tag{18}$$

Throughout we use $\gamma = 0.4$ in (7).

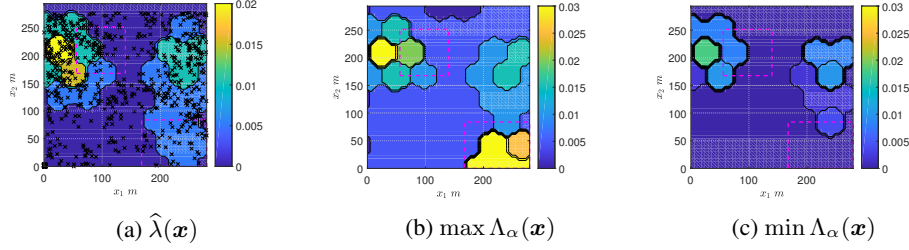

(a) $\widehat{\lambda}(\boldsymbol{x})$      (b) $\max \Lambda_\alpha(\boldsymbol{x})$      (c) $\min \Lambda_\alpha(\boldsymbol{x})$

Figure 3: # trees per m$^2$. Nominal coverage set to $1 - \alpha = 80\%$. The dashed boxes mark missing data regions.

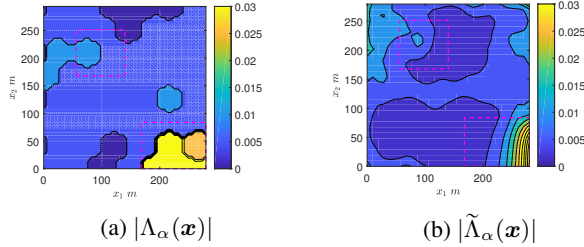

(a) $|\Lambda_\alpha(\boldsymbol{x})|$      (b) $|\widetilde{\Lambda}_\alpha(\boldsymbol{x})|$

Figure 4: # trees per m$^2$. Comparison between proposed intensity interval and credibility intensity interval from approximate posterior of LGCP model.

**Hickory tree data**

First, we consider the hickory trees data set [1] which consists of coordinates of hickory trees in a spatial domain $\mathcal{X} \subset \mathbb{R}^2$, shown in Figure 3a, that is equipartitioned into a regular lattice of $R = 52$ hexagonal regions. The dataset $(\boldsymbol{r}, \boldsymbol{y})$ contains the observed number of trees in each region. The dashed boxes indicate regions data inside which is considered to be missing. For the cubic b-splines in $\mathcal{P}_{\boldsymbol{\theta}}$, the spatial support was again set to cover all regions.

We observe that the point predictor $\widehat{\lambda}(\boldsymbol{x})$ interpolates and extrapolates smoothly across regions and appears to visually conform to the density of the point data. Figures 3b and 3c provide important complementary information using $\Lambda_\alpha(\boldsymbol{x})$, whose upper limit increases in the missing data regions, especially when extrapolating in the bottom-right corner, and lower limit rises in the dense regions.

The size of the interval $|\Lambda_\alpha(\boldsymbol{x})|$ quantifies the predictive uncertainty and we compare it to the $(1-\alpha)\%$ credibility interval $|\widetilde{\Lambda}_\alpha(\boldsymbol{x})|$ using the LGCP model as above, cf. Figures 4a and 4b. We note that the sizes increase in different ways for the missing data regions. For the top missing data region, $|\widetilde{\Lambda}_\alpha(\boldsymbol{x})|$ is virtually unchanged in contrast to $|\Lambda_\alpha(\boldsymbol{x})|$. While $|\widetilde{\Lambda}_\alpha(\boldsymbol{x})|$ appears relatively tighter than $|\Lambda_\alpha(\boldsymbol{x})|$ across the bottom-right missing data regions, the credible interval lacks any out-of-sample guarantees that would make the prediction reliable.

**Crime data**

Next we consider crime data in Portland police districts [16, 10] which consists of locations of calls-of-service received by Portland Police between January and March 2017 (see figure 5a). The spatial region $\mathcal{X} \subset \mathbb{R}^2$ is equipartitioned into a regular lattice of $R = 494$ hexagonal regions. The dataset $(\boldsymbol{r}, \boldsymbol{y})$ contains the reported number of crimes in each region. The support of the cubic b-spline is taken to be 12 km.

The point prediction $\widehat{\lambda}(\boldsymbol{x})$ is shown in Figure 5a, while Figures 5b and 5c plot the upper and lower limits of $\Lambda_\alpha(\boldsymbol{x})$, respectively. We observe that $\widehat{\lambda}(\boldsymbol{x})$ follows the density of the point pattern well, predicting a high intensity of approximately 60 crimes/km$^2$ in the center. Moreover, upper and lower limits of $\Lambda_\alpha(\boldsymbol{x})$ are both high where point data is dense. The interval tends to being noninformative for regions far away from those with observed data, as is visible in the top-left corner when comparing Figures 5b and 5c.

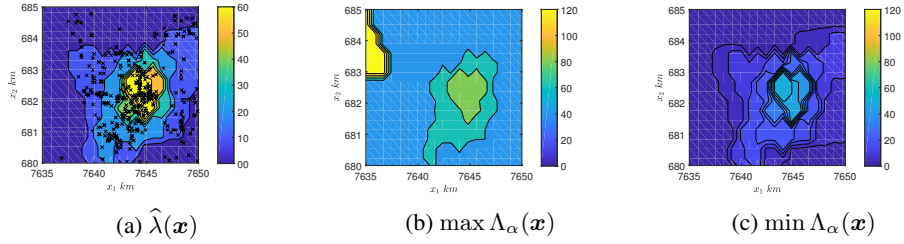

(a) $\widehat{\lambda}(\boldsymbol{x})$      (b) $\max \Lambda_\alpha(\boldsymbol{x})$      (c) $\min \Lambda_\alpha(\boldsymbol{x})$

Figure 5: # crimes per km$^2$ in Portland, USA. Nominal coverage set to $1 - \alpha = 80\%$.

## 5   Conclusion

We have proposed a method for inferring predictive intensity intervals for spatial point processes. The method utilizes a spatial Poisson model with an out-of-sample accuracy guarantee and the resulting interval has an out-of-sample coverage guarantee. Both properties hold even when the model is misspecified. The intensity intervals provide a reliable and informative measure of uncertainty of the point process. Its size is small in regions with observed data and grows along missing regions further away from data. The proposed regularized learning criterion also leads to more informative intervals as compared to an unregularized maximum likelihood approach, while its statistical guarantees renders it reliable in a wider range of conditions than standard methods such as LGCP inference. The method was demonstrated using both real and synthetic data.

### Acknowledgments

The work was supported by the Swedish Research Council (contract numbers $2017 - 04610$ and $2018 - 05040$).

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
