[Supplementary Material · supp.pdf]

# Supplementary Material:
**Prediction of Spatial Point Processes:**
**Regularized Method with Out-of-Sample Guarantees**

## Spatial basis $\phi(r)$

For space $\mathcal{X}$ divided into $R$ regions with each region $\mathcal{X}_r$ denoted by its region index $r$, the spatial basis vector evaluated at $r$ is an $R \times 1$ vector given by

$$\phi(r) = \text{col}\{\phi_1(r), \ldots, \phi_R(r)\}.$$

Here $\phi(r)$ is the cubic b-spline in space with two parameters: center and support. The $k^{th}$ component i.e $\phi_k(r)$ has its center at the center of region $\mathcal{X}_k$ and peak value when evaluated at $k$. The support of $\phi_k(r)$ determines its value at the neighbouring regions and hence allows to control the local structure in intensity in our model. For details on cubic b-spline see [2].

## Dual Norm $\widetilde{f}(\cdot)$

Let $f(\boldsymbol{\theta}) = ||\boldsymbol{w} \odot \boldsymbol{\theta}||_1$. By definition of dual norm,

$$\widetilde{f}(\boldsymbol{g}) = \sup_{\boldsymbol{\theta}:f(\boldsymbol{\theta})\leqslant 1} \boldsymbol{g}^{\top}\boldsymbol{\theta}$$

The condition $f(\boldsymbol{\theta}) \leqslant 1$ implies

$$\sum_{k=1}^{R} |w_k||\theta_k| \leqslant 1, \quad \min_{k=1,\ldots,R}|w_k| \sum_{k=1}^{R}|\theta_k| \leqslant 1, \quad ||\boldsymbol{\theta}||_1 \leqslant w_o^{-1},$$

where $w_o = \min_{k=1,\ldots,R}|w_k|$. Moreover,

$$\boldsymbol{g}^{\top}\boldsymbol{\theta} = \sum_{i=1}^{R} g_k\theta_k \leqslant \sum_{i=1}^{R}|g_k||\theta_k| \leqslant ||\boldsymbol{g}||_\infty ||\boldsymbol{\theta}||_1$$

Combining this with $||\boldsymbol{\theta}||_1 \leqslant w_o^{-1}$ we get

$$\widetilde{f}(\boldsymbol{g}) = \frac{||\boldsymbol{g}||_\infty}{w_o}$$

## Hoeffding's inequality for $z_k$

We show that $z_k$ is bounded in $[-Y, Y]$ and hence we can make use of Hoeffding's inequality to get eq. (14).

The gradient of eq. (10) evaluated at $\widehat{\boldsymbol{\theta}}$ is

$$\boldsymbol{g} = \left[\mathbb{E}_n[z_1], \ldots, \mathbb{E}_n[z_R]\right]^{\top},$$

where $z_k = (y - \mathbb{E}_{y|r}[y])\phi_k(r)$. Given that the maximum number of counts is bounded i.e. $\max y \leqslant Y$, we have

$$\max z_k = \max\left\{(y - \mathbb{E}_{y|r}[y])\phi_k(r)\right\} = \max\{(y - \mathbb{E}_{y|r}[y])\}\max\{\phi_k(r)\} = Y,$$
$$\min z_k = \min\left\{(y - \mathbb{E}_{y|r}[y])\phi_k(r)\right\} = \min\{(y - \mathbb{E}_{y|r}[y])\}\max\{\phi_k(r)\} = -Y,$$

for all $k = 1, \ldots, R$. Here $\max \phi_k(r) = 1$.

## Union bound and DeMorgan's Law

Given that $\mathbb{E}[z_k] = 0$, from eq. (14) we get

$$\Pr(|\mathbb{E}_n[z_k]| \leqslant \epsilon) \geqslant 1 - 2\exp\left[-\frac{n\epsilon^2}{2Y^2}\right].$$

Moreover,

$$\Pr\left(\max_{k=1,\ldots,R}|\mathbb{E}_n[z_k]| \leqslant \epsilon\right) = \Pr\left(\bigcap_{k=1}^{R}|\mathbb{E}_n[z_k]| \leqslant \epsilon\right).$$

By DeMorgan's law,

$$\Pr\left(\bigcap_{k=1}^{R}|\mathbb{E}_n[z_k]| \leqslant \epsilon\right) = \Pr\left(\bigcup_{k=1}^{R}|\mathbb{E}_n[z_k]| \geqslant \epsilon\right)'.$$

By union bound,

$$\Pr\left(\bigcup_{k=1}^{R}|\mathbb{E}_n[z_k]| \geqslant \epsilon\right) \leqslant \sum_{i=1}^{R}\Pr\left(|\mathbb{E}_n[z_k]| \geqslant \epsilon\right) = 2R\exp\left[-\frac{n\epsilon^2}{2Y^2}\right],$$

which implies that

$$\Pr\left(\bigcap_{k=1}^{R}|\mathbb{E}_n[z_k]| \leqslant \epsilon\right) \geqslant 1 - 2R\exp\left[-\frac{n\epsilon^2}{2Y^2}\right].$$

Eq. (15) follows from above.

## Minimization Algorithm

Here we derive the majorization-minimization (MM) algorithm that is used to solve eq. (7). Let $V(\boldsymbol{\theta}) = -n^{-1}\ln p_{\boldsymbol{\theta}}(\boldsymbol{y}|\boldsymbol{r})$ and $f(\boldsymbol{\theta}) = ||\boldsymbol{w} \odot \boldsymbol{\theta}||_1$. For the Poisson model class considered in the paper,

$$V(\boldsymbol{\theta}) = n^{-1}\left(\sum_{i=1}^{n}\mathbb{E}_{\boldsymbol{\theta}}[y_i|r_i] - y_i\ln(\mathbb{E}_{\boldsymbol{\theta}}[y_i|r_i]) + \ln(y_i!)\right),$$

where $\mathbb{E}_{\boldsymbol{\theta}}[y_i|r_i] = \exp(\boldsymbol{\phi}(r_i)^{\top}\boldsymbol{\theta})$. $V(\boldsymbol{\theta})$ is convex in $\boldsymbol{\theta}$ since

$$\partial_{\boldsymbol{\theta}}^2 V(\boldsymbol{\theta}) = n^{-1}\boldsymbol{\Phi}\mathbf{D}\boldsymbol{\Phi}^{\top} \geq 0.$$

Here,
$$\boldsymbol{\Phi} \;=\; [\boldsymbol{\phi}(r_1), \ldots, \boldsymbol{\phi}(r_R)] \quad and \quad \mathbf{D} \;=\; diag(\boldsymbol{h}(\boldsymbol{\theta}))$$
are an $R \times R$ basis and diagonal matrices respectively and
$$\boldsymbol{h}(\boldsymbol{\theta}) \;=\; \mathrm{col}\{\mathbb{E}_{\boldsymbol{\theta}}[y_1|r_1], \ldots, \mathbb{E}_{\boldsymbol{\theta}}[y_R|r_R]\}.$$

By convexity of $V(\boldsymbol{\theta})$, given an initial estimate $\widetilde{\boldsymbol{\theta}}$, the objective in eq. (7) can be upper bounded as
$$V(\boldsymbol{\theta}) + n^{-\gamma}f(\boldsymbol{\theta}) \leqslant Q(\boldsymbol{\theta}; \widetilde{\boldsymbol{\theta}}) + n^{-\gamma}f(\boldsymbol{\theta}),$$

where $Q(\boldsymbol{\theta}; \widetilde{\boldsymbol{\theta}})$ is a quadratic majorization function (see [1], ch. 5) of $V(\boldsymbol{\theta})$ given by
$$Q(\boldsymbol{\theta}; \widetilde{\boldsymbol{\theta}}) = V(\widetilde{\boldsymbol{\theta}}) + \boldsymbol{v}^{\top}(\boldsymbol{\theta} - \widetilde{\boldsymbol{\theta}}) + \frac{1}{2}(\boldsymbol{\theta} - \widetilde{\boldsymbol{\theta}})^{\top}\mathbf{H}(\boldsymbol{\theta} - \widetilde{\boldsymbol{\theta}}).$$

Here $\boldsymbol{v} \;=\; \partial_{\boldsymbol{\theta}}V(\boldsymbol{\theta})|_{\boldsymbol{\theta} \,=\, \widetilde{\boldsymbol{\theta}}} \;=\; n^{-1}\boldsymbol{\Phi}(\boldsymbol{h}(\widehat{\boldsymbol{\theta}}) - \boldsymbol{y})$ and $\mathbf{H} \;=\; \max_{\boldsymbol{\theta}}\{\partial_{\boldsymbol{\theta}}^2 V(\boldsymbol{\theta})\}$. The diagonal elements of $\mathbf{D}$ represent the average number of counts in different regions. Given that the counts in any region are bounded i.e. $y \leqslant Y$, $\mathbf{H} \preceq n^{-1}Y\boldsymbol{\Phi}\boldsymbol{\Phi}^{\top}$ therefore we have

$$V(\boldsymbol{\theta}) + n^{-\gamma}f(\boldsymbol{\theta}) \leqslant V(\widehat{\boldsymbol{\theta}}) + n^{-1}(\boldsymbol{h}(\widehat{\boldsymbol{\theta}}) - \boldsymbol{y})^{\top}\boldsymbol{\Phi}^{\top}(\boldsymbol{\theta} - \widetilde{\boldsymbol{\theta}}) + \frac{Y}{2n}||\boldsymbol{\Phi}^{\top}(\boldsymbol{\theta} - \widetilde{\boldsymbol{\theta}})||_2^2 + n^{-\gamma}f(\boldsymbol{\theta}). \tag{SM1}$$

Therefore starting from an initial estimate $\widetilde{\boldsymbol{\theta}}$, one can minimize the right hand side of (SM1) to obtain $\check{\boldsymbol{\theta}}$ then update $\widetilde{\boldsymbol{\theta}} \;=\; \check{\boldsymbol{\theta}}$ and repeat until convergence to get final solution of eq. (7) $\widehat{\boldsymbol{\theta}} \;=\; \check{\boldsymbol{\theta}}$. The pseudocode is given in algorithm (2).

Furthermore, the right hand side of (SM1) can be transformed into a weighted lasso regression problem and hence can be efficiently solved using coordinate descent algorithm [1]. Letting $\boldsymbol{q}(\widetilde{\boldsymbol{\theta}}) \;=\; \boldsymbol{\Phi}^{\top}\widetilde{\boldsymbol{\theta}} \;+\; Y(\boldsymbol{y} - \boldsymbol{h}(\widetilde{\boldsymbol{\theta}}))$, the right hand side of (SM1) can be rewritten as

$$Y(2n)^{-1}(\boldsymbol{q}(\widetilde{\boldsymbol{\theta}}) - \boldsymbol{\Phi}^{\top}\boldsymbol{\theta})^{\top}(\boldsymbol{q}(\widetilde{\boldsymbol{\theta}}) - \boldsymbol{\Phi}^{\top}\boldsymbol{\theta}) + n^{-\gamma}f(\boldsymbol{\theta}) + K(\widetilde{\boldsymbol{\theta}}),$$

where the first two terms form a weighted lasso regression problem in $\boldsymbol{\theta}$ and the last term $K(\widetilde{\boldsymbol{\theta}}) \;=\; V(\widetilde{\boldsymbol{\theta}}) - \boldsymbol{q}(\widetilde{\boldsymbol{\theta}})$ is independent of $\boldsymbol{\theta}$ and does not affect the minimization problem. This conclude the derivation of the MM algorithm.