[Reviews · NeurIPS 2019]

Reviewer 1



The paper under review introduces a new inference methodology for the estimation of spatial point processes intensity for which the authors derive confidence interval with guaranteed out-of-sample performance. The approach combines a discretised Poisson process with the conformal prediction framework to obtain point-wise confidence intervals. The authors propose an efficient inference procedure based on a majorization-minimization algorithm. The approach is illustrated by simulations and applied to two different data sets. The paper is very well written but some imprecisions in the notations make the exposition sometime difficult to follow, for instance it is a good practice to denote random variables with upper case letters. This work is original and of good quality, the reviewer has the following comments: Minor points: - l.3-4: the authors mention a tuning-free regularized criterion, but the choice of $\gamma$ is never discussed and chosen arbitrarily. Remove this point or give further explanations. - l.27: `a spatially varying intensity interval' the formulation is awkward. Maybe, develop `an inference procedure for the spatial intensity of a point process'. Inference imply directly that you derive not only a point estimate but also confidence intervals. - l.29-35: the choice a bullet list is strange as only the first bullet is a contribution, the other three are precisions/details on this contribution. - l.39: NotationS - Figure 1 is really misleading as it shows an example where the estimator does not rely on discretization of the space, which make the presentation confusing. Should be changed. - l.44: the index $r$ is missing the union. Isn't $\lambda(x)$ the expected number of event? Is $y$ an event or the number of events? $Y$ is not defined (maximum number of events?). - l.47: Further details about the motivation for partitioning should be given (more than `it is usual practice'). - Equation (3): I don't really understand the notation. Shouldn't be $Pr\{y \in \Lambda(x) \} > 1 - \alpha$ for all $x \in \chi$ as suggested by Algorithm 1? - l.64: $r$ is an index and not a region. - l.65: what do you mean by `free to vary $\tilde{y}$'? - Algorithm 1: In 3, shouldn't be find the $r$ corresponding to $x$? In 5, do you accumulate the score the ones of previous value of $\tilde{y}$? - Equation 5: in KL, there is no $n$. The empirical estimator of the KL should be introduced later. Or replace be the log-likelihood function ($p(y|r)$ is not accessible in practice). - l.88-89: $\Phi$ is not defined in the equation. - Equation 6: now it should be the log-likelihood function. - Equation 7: $\gamma$ is not defined. - l.102: why do you restrict $\gamma$ to $[(0,1/2)$? Can you comment on it? - l.107+: `we then obtain..` is it the estimator you obtain with this particular choice of weights? Then the new $\hat{\theta}$ might benefit from a new notation to stress the difference. - l.110: $p(y|r)$ is not known and it is not possible to compute the empirical divergence. - Algorithm 2: why not using `while' and specify the exact criteria for convergence? - Section 4.1: Might be worth mentioning that $Y = 10$. - Section 4.1: would it be possible to have (in Appendix) the empirical coverage rate? As this is a simulation study, that can be easily computed and would be highly pertinent. Indeed, your methodology might have reach $100\%$ easily (too conservative) while the likelihood approach might have a lower but still good coverage. Those tables would help to understand the differences. - Figure 2. a): Can you explain why the likelihood approach yields wider confidence interval on the left side on the interval? This sounds conter-intuitive as your method ought to be more conservative. - Figure 2. b): The bad performance of the likelihood seems caused here by a not strong enough regularization/bad choice of priori. - Section 4.2: what is the value of $Y$ and how did you choose it? Does it impact the results? - l. 204-205: from the simulation study we cannot say that your methodology yields more informative intervals. Indeed an interval with $100\%$ coverage is not informative. So either you give the empirical coverage numbers of the simulation study to illustrate that it is the case in this setting, or you just re-formulate with `leads to intervals with guaranteed out-of-sample coverage level'.

Reviewer 2



This paper presents a regularized method of spatial point process to infer predictive intensity intervals. The intensity interval is constructed using a spatial Poisson model with provable out of sample accuracy. The method is demonstrated using both synthetic and real spatial data. In this work, the intensity interval is developed using the conformal prediction framework. The intensity interval exhibits provable out of sample prediction performance guarantee. My major concern is the practicality of the proposed method. For the intensity interval where there is missing data, the estimator is relatively useful in the interpolated area. It is not surprising that the intensity interval grows drastically in the extrapolated area (see the right side of Figure 1). Also regarding the learning criterion (7), the first term (log-likelihood) is proportional to n^-1, and the second term (regularization) is proportional to n^-gamma, where gamma is close to 0.5. Hence the two terms look a bit imbalanced. It will be helpful if the proposed method can be further tested with more datasets and clarified in details. Also, it will be helpful if the proposed method can be compared with the state of the art methods besides LCGP. Moreover, it will be useful to discuss the computation complexity of the proposed method.

Reviewer 3



I think the application of conformal prediction to spatial point processes within the ML community is interesting, particularly as most recent work in this area has taken on a Bayesian flavor, and we lack sufficient rigor in evaluating accuracy of uncertainty quantification. I do think that it could be made a little more clear that the intervals require a partitioning of the domain, and I would be interested to understand how this partitioning affects the performance (and bound)---is improper discretization covered under the model-misspecification guarantee? In general, I think the submission is reasonably well written, novel (to the best of my knowledge), and would be of interest to those in the community who work on point processes.

[Author Response · NeurIPS 2019]

**Reviewer 2**: Regarding comments on parameter $\gamma > 0$: Firstly, the condition $\gamma \in (0, \ 0.5)$ is a sufficient condition for the bound in Theorem 1 to hold, which follows from the proof (see eq. (15)). Secondly, if $\gamma$ is increased, the bound tightens but on the other hand the probabilistic guarantee weakens. However, for a given data set one can readily search for the $\gamma$ in $(0, \ 0.5)$ which yields the most informative (tightest) interval $\Lambda_\alpha(\boldsymbol{x})$. In the revised paper, we will elaborate on these points and amend the wording of 'tuning-free'.

Regarding comments on Fig. 1: The spatial domain in this example is in fact discretized into $R = 50$ regions whose resulting small size only make the curves *appear* continuous. The revised paper clarifies this.

Regarding comments related to the risk $\mathcal{R}(\boldsymbol{\theta})$: First, note that in eq. (5) the risk can also be writtten as $\mathcal{R}(\boldsymbol{\theta}) = -n^{-1}\mathbb{E}_{\boldsymbol{y}|\boldsymbol{r}}[\ln p_{\boldsymbol{\theta}}(\boldsymbol{y}|\boldsymbol{r})] + K$, where the constant $K = n^{-1}\mathbb{E}_{\boldsymbol{y}|\boldsymbol{r}}[\ln p(\boldsymbol{y}|\boldsymbol{r})]$ is there only to ensure nonnegativity $\mathcal{R}(\boldsymbol{\theta}) \geq 0$ and the natural property $\mathcal{R}(\boldsymbol{\theta}) = 0 \Leftrightarrow p_{\boldsymbol{\theta}}(\boldsymbol{y}|\boldsymbol{r}) \equiv p(\boldsymbol{y}|\boldsymbol{r})$. Thus including $K$ in the definition of risk is natural and it requires no knowledge of $p(\boldsymbol{y}|\boldsymbol{r})$ in order to define the unknown $\boldsymbol{\theta}^\star$ in eq. (6) as it does not affect the optimization problem. Secondly, since $\ln p(\boldsymbol{y}|\boldsymbol{r}) = \sum_{i=1}^{n} \ln p(y_i|r_i)$ and $\ln p_{\boldsymbol{\theta}}(\boldsymbol{y}|\boldsymbol{r}) = \sum_{i=1}^{n} \ln p_{\boldsymbol{\theta}}(y_i|r_i)$ are a sum of $n$ terms and division by $n$ in eq. (5) is merely a natural normalization to make derivation and final expressions neater. The risk $\mathcal{R}(\boldsymbol{\theta})$ is therefore the Kullback-Leibler divergence *per sample*. We will clarify this is the revised manuscript.

Regarding comments related to empirical coverage: Let $\Lambda_\alpha(\boldsymbol{x})$ and $\Lambda_\alpha^{'}(\boldsymbol{x})$ denote the prediction intervals for the proposed regularized approach (in eq. (7)) and the standard likelihood approach, respectively. In Section 4.1, we conduct a small but illustrative simulation study in which both intervals are computed using the conformal prediction framework (Algorithm 1). Both intervals have empirical coverages that are valid, that is, exceed $1 - \alpha = 80\%$. The difference between the intervals lies in their sizes as shown in Fig. 2c, where $\Lambda_\alpha(\boldsymbol{x})$ is found to be much smaller than $\Lambda_\alpha^{'}(\boldsymbol{x})$ for the same coverage. In the interest of more details, we will include additional simulations in the supplementary material that evaluate the empirical coverages of the two intervals under different scenarios.

Regarding comments related to notational issues: $r$ has now been added in union in line $44$. The intensity function $\lambda(\boldsymbol{x})$ is defined as the number of events *per unit area*. This has now been clarified on line $44$. Capital $Y$ is the maximum number of events in a region and only plays the role of a practical upper limit on the conformal prediction interval so as to terminate the for-loop in Algorithm 1. Given that it is set reasonably high for the problem at hand, it does not affect the results and in practice we simply set it to an integer multiple of the maximum number of counts observed in any region. Details regarding $\phi(r)$ have now been added on lines $89 - 90$. In Algorithm 2, **'repeat'** has now been replaced by **'while'** and the convergence criteria specified. Other notational issues have also been addressed and will appear in the revised paper.

**Reviewer 3**: Regarding comments on utility of the method in extrapolation scenarios: Unless a method is based on some plausible physical model of the point process, one cannot expect it to yield valid *and* informative intervals far from the observed data. Indeed the proposed method yields valid intervals but they become increasingly uninformative for regions further from the observed data. By contrast, the standard kernel density-based methods (e.g. [6]) may wrongly infer the absence of events outside data as zero intensity in this case, whereas the credible intervals obtained from Bayesian method become less informative but do not exhibit any statistical validity (see Fig. 2b). Hence proposed method yields intervals that reflect uncertainty due to missing data in a statistically appropriate way (see Fig. 2a and 2b).

Regarding comments on the balance of the terms in the fitting criterion in eq. (7): Note that $-n^{-1} \ln p_{\boldsymbol{\theta}}(\boldsymbol{y}|\boldsymbol{r}) = -n^{-1} \sum_{i=1}^{n} \ln p_{\boldsymbol{\theta}}(y_i|r_i)$ is a sum of $n$ terms therefore it will not decay as the number of datapoints $n$ increase. On the other hand, the second regularization term will decay with $n$, as desired. Thus the fitting criterion in eq. (7) is appropriately balanced.

Regarding comments on comparison with other methods: In the literature, the Cox process is the default model and state-of-the-art methods are based using log-Gaussian link function (LGCP in Section 4.1). Given the intractibility of this model, most work is centered on computational approximations ([9], [13], [20], [2]). In this work, we have compared our method with the popular integrated Laplace approximation (INLA [17], [11]). Less tractable Monte Carlo approximations, e.g. [9], [2] were not implemented in this study.

Regarding comments on the computational complexity of the proposed method: Algorithm 2 solves a series of weighted lasso problems and can therefore be solved in a runtime that scales as $\mathcal{O}(nR^2)$ where $n$ is the number of datapoints and $R$ is the dimension of $\boldsymbol{\theta}$. This has been clarified in the revised paper.

**Reviewer 4**: Regarding comments related to effect of discretization: the manner in which space is discretized will affect the size of intervals but not their statistical validity. That is, irrespective of how space is discretized, the out-of-sample guarantees eqs. (3) and (8) will still hold, whether the resulting model is misspecified or not. We will elaborate on the effect of spatial discretization in the revised paper.

[Meta-Review · NeurIPS 2019]

Congratulations, your paper has been accepted for publication at NeurIPS2019. The extension of the conformal prediction framework to point processes is novel and interesting, and I think it will make a nice contribution to the conference. Conformal prediction is not widely understood within ML, and so I hope this paper adds to the discussion around appropriate inferential frameworks. When preparing the camera ready version, please bear in mind the reviewers' comments. In particular, the following were raised during the discussion as outstanding points that should be considered when revising the paper: - consider the comments about improving the notation to make the exposition clearer. In particular, I found the definition of r in eq (2) confusing at first. - please include the additional results on the empirical coverages for the simulation study presented in the supplementary material. Without these results, the reader cannot tell if the confidence intervals are informative or just guaranteed. - clarify whether improper discretization is covered under the model-misspecification guarantee?